# Differentiable Neuro-Symbolic Reasoning on Large-Scale Knowledge Graphs

**Shengyuan Chen**
Department of Computing
The Hong Kong Polytechnic University
Hung Hom, Hong Kong SAR
shengyuan.chen@connect.polyu.hk

**Yunfeng Cai**
Cognitive Computing Lab
Baidu Research
10 Xibeiwang East Rd., Beijing, China
caiyunfeng@baidu.com

**Huang Fang**
Cognitive Computing Lab
Baidu Research
10 Xibeiwang East Rd., Beijing, China
fanghuang@baidu.com

**Xiao Huang**
Department of Computing
The Hong Kong Polytechnic University
Hung Hom, Hong Kong SAR
xiaohuang@comp.polyu.edu.hk

**Mingming Sun**
Cognitive Computing Lab
Baidu Research
10 Xibeiwang East Rd., Beijing, China
sunmingming01@baidu.com

## Abstract

Knowledge graph (KG) reasoning utilizes two primary techniques, i.e., rule-based and KG-embedding based. The former provides precise inferences, but inferring via concrete rules is not scalable. The latter enables efficient reasoning at the cost of ambiguous inference accuracy. Neuro-symbolic reasoning seeks to amalgamate the advantages of both techniques. The crux of this approach is replacing the predicted existence of all possible triples (i.e., truth scores inferred from rules) with a suitable approximation grounded in embedding representations. However, constructing an effective approximation of all possible triples' truth scores is a challenging task, because it needs to balance the tradeoff between accuracy and efficiency, while compatible with both the rule-based and KG-embedding models. To this end, we proposed a differentiable framework - DiffLogic. Instead of directly approximating all possible triples, we design a tailored filter to adaptively select essential triples based on the dynamic rules and weights. The truth scores assessed by KG-embedding are continuous, so we employ a continuous Markov logic network named probabilistic soft logic (PSL). It employs the truth scores of essential triples to assess the overall agreement among rules, weights, and observed triples. PSL enables end-to-end differentiable optimization, so we can alternately update embedding and weighted rules. On benchmark datasets, we empirically show that DiffLogic surpasses baselines in both effectiveness and efficiency.

## 1 Introduction

Knowledge graph (KG) reasoning refers to the process of using existing triples in a KG to infer new knowledge that is not explicitly stated in the original KG (Safavi & Koutra, 2020). For instance, if we know that *(Paris, is_the_capital_of, France)* and *(France, is_a_country_in, Europe)*, we can infer

37th Conference on Neural Information Processing Systems (NeurIPS 2023).

that *(Paris, is_a_city_in, Europe)*, which is not explicitly included in the original KG. KG reasoning is essential to KG completion (Shi & Weninger, 2017) and KG error detection (Zhang et al., 2022a; Dong et al., 2023b). Moreover, KG reasoning can infer underlying knowledge and improve the quality of learning or predictions on KGs, which benefits various downstream tasks, such as question answering (Huang et al., 2019; Dong et al., 2023a), recommendations (Huang et al., 2023; Xu et al., 2023), and interpretable machine learning (Lecue, 2020; Tiddi & Schlobach, 2022).

There are mainly two lines of research in KG reasoning (Zhang et al., 2022b). First, rule-based reasoning derives new triples from existing ones by applying a set of predefined rules, which are usually expressed in a form of logical statements (Yang et al., 2017; Sadeghian et al., 2019; Fang et al., 2023). For example, if there are two triples *(A, is_a_parent_of, B)* and *(B, is_a_parent_of, C)*, and a rule saying "the parent of a parent is a grandparent", then we can infer a new triple *(A, is_a_grandparent_of, C)*. Rule-based reasoning is particularly effective when the rules are well-defined and the KG is relatively small and static (Bach et al., 2017). Recent studies employ Markov logic network (MLN) (Richardson & Domingos, 2006) to dynamically learn soft rules, which are associated with weight scores indicating their credibility. But it remains not scalable because assessing the truth scores to all possible triples requires $\mathcal{O}(|\mathcal{E}|^2|\mathcal{R}|)$ parameters, where $|\mathcal{E}|$ and $|\mathcal{R}|$ denote the numbers of entities and relations. Thus, rule-based reasoning becomes inefficient when dealing with large KGs. Second, KG-embedding based reasoning projects a KG into a low-dimensional space and infers new relations based on embedding representations (Zhang et al., 2021; Ren et al., 2022). It assumes that KG embedding can preserve most semantic information in the original KGs (Bordes et al., 2013; Wang et al., 2014; Sun et al., 2019; Dong et al., 2014) so that missing relations can be inferred by using the distances or semantic matching scores in the embedding space. Recent studies (Shang et al., 2019; **?**; Vashishth et al., 2019; Zhang et al., 2023) also seek to design tailored Graph convolutional networks for learning structural-aware KG embeddings. KG-embedding based reasoning can be scalable (Zheng et al., 2020; Ren et al., 2022). However, there is no explicit rule to ensure its accuracy and logical consistency. Also, it requires a sufficient amount of training data, and its results are difficult to interpret.

Neuro-symbolic reasoning models combine the advantages of both techniques, which is to approximate the predicted truth scores of all possible triples inferred by rules with the normalized output scores of a KG-embedding model. However, it is nontrivial to effectively and efficiently construct such an approximation. Directly employing rules (Guo et al., 2016, 2018) to regularize the embedding learning is efficient, but it cannot update rule weights and is thus sensitive to the initialization of rule weights. Incorporating KG-embedding models into sophisticated rule-based models, such as MLN, enables the handling of rule uncertainty (Qu & Tang, 2019; Zhang et al., 2020). However, directly approximating the distributions of the truth scores are still challenging. First, directly approximating the distribution of truth scores within the MLN framework is unfeasible. This process necessitates the optimization of the joint probability of the approximated distribution, which subsequently requires the computation of integration across all scores. Second, optimizing neuro-symbolic models rely on training with ground formulas (i.e. instantiated rules). Given the large grounding space, the performance of neuro-symbolic models may deteriorate if important ground formulas are missed (Guo et al., 2016; Qu & Tang, 2019), and efficiency issues may arise if too many ground formulas are considered (Zhang et al., 2020).

To this end, we proposed a differentiable framework - DiffLogic. Firstly, a tailored filter is used to adaptively select important ground formulas based on weighted rules and extract connected triples. Second, a KG-embedding model is used to compute a truth score for each triple. Then we employ a continuous MLN named probabilistic soft logic (PSL) (Bach et al., 2017) that takes these truth scores as input, and assesses the overall agreement among rules, weights, and observed triples with a joint probability. The PSL template enables end-to-end differentiable optimization. In this way, DiffLogic is optimized by alternately updating embedding and weighted rules. Contributions of this work are summarized as follows:

(1) We develop a unified neuro-symbolic framework — DiffLogic, that combines the advantages of KG-embedding models and rule-based models: efficiency, effectiveness, capability of leveraging prior knowledge and handling uncertainty.

(2) We enable consistent training of rule-based models and KG-embedding models. By employing PSL, the joint probability of truth scores can be optimized directly rather than optimizing an evidence lower bound (ELBO) instead.

(3) We propose an efficient grounding technique that iteratively identifies important ground formulas required for inference, enabling effective and data-efficient optimization.

(4) We devise a fast estimation technique for the gradient of rule weights, which efficiently estimates the rule weight gradient by exploiting the sparsity of violated ground formulas.

(5) Through experiments, we empirically show that: (i) DiffLogic scales to large knowledge graphs with consistently improved performance, such as YAGO3-10; (ii) our model can leverage human prior knowledge by injecting rule patterns using only a compact set of logic rules; and (iii) our grounding technique is both efficient and effective, it reduces the number of ground formulas required for optimization by orders of $10^3 \sim 10^5$, without compromising performance.

## 2 Preliminaries

In this section, we formally define the problem of knowledge base completion, provide a brief introduction to first-order logic, and how to evaluate its agreement with a knowledge base.

### 2.1 Problem statement

A knowledge base $\mathcal{K}$ comprises a set of entities $\mathcal{E}$ and a set of relations $\mathcal{R}$. For any pair of head-tail entities $(h, t) \in \mathcal{E} \times \mathcal{E}$ and a relation $r \in \mathcal{R}$, the relation maps the pair of entities to a binary score, i.e., $r : \mathcal{E} \times \mathcal{E} \to \{0, 1\}, \forall r \in \mathcal{R}$, indicating that the head entity $h$ either has the relation $r$ with the tail entity $t$ or not. In the knowledge base completion problem, people observe a set of facts $\mathcal{O} = \{(h_i, r_i, t_i)\}_{i=1}^n$ along with their true assignment vector $\boldsymbol{x} = [x_1, \ldots, x_{|\mathcal{O}|}] \in \mathbb{R}^{|\mathcal{O}|}$ with $x_i = r_i(h_i, t_i)$. Denote the unobserved facts as $\mathcal{H} = \mathcal{E} \times \mathcal{R} \times \mathcal{E} \backslash \mathcal{O}$, and let $\{(F_q, W_q)\}_{q=1}^m$ be a set of weighted rules, where $F_q$ is a rule (see Section 2.2 for details, where a rule is referred to as first-order logic), $W_q$ is the corresponding rule weight. The knowledge graph completion task aims to infer the assignment vector for all unobserved facts $\boldsymbol{y} \in \mathbb{R}^{|\mathcal{H}|}$ given the observed facts and the rules.

### 2.2 First-order logic

*First-order logic.* A first-order logic (also referred to as "logic rule" in this paper) associated with a knowledge base $\mathcal{K}$ is an expression based on relations in $\mathcal{K}$. Formally, a logic rule $F_q$ in clausal form can be represented as a disjunction of atoms and negated atoms:

$$\left( \vee_{i \in \mathcal{I}_q^+} r_i(A_i, B_i) \right) \vee \left( \vee_{i \in \mathcal{I}_q^-} \neg r_i(A_i, B_i) \right), \tag{1}$$

where $\mathcal{I}_q^-$ and $\mathcal{I}_q^+$ are two index sets containing the indices of atoms that are negated or not, respectively, $A_i$ and $B_i$ are variables. Logic rules in clausal form can be equivalently reorganized as an implication from premises (negated) to conclusions (non-negated):

$$\wedge_{i \in \mathcal{I}_q^-} r_i(A_i, B_i) \implies \vee_{i \in \mathcal{I}_q^+} r_i(A_i, B_i). \tag{2}$$

The implication Eq. (2) is quite expressive since it includes many commonly used types of logic rule, e.g., symmetry/asymmetry, inversion, sub-relation, composition, etc.

### 2.3 Rule grounding and distance to satisfaction

*Rule grounding.* For a logic rule $F_q$ in Eq. (2), by assigning entities $e \in \mathcal{E}$ to the variables $A_i$ and $B_i$, $F_q$ is *grounded*, producing a set of *ground formulas*. For example, consider a simple logic rule $Father(A, B) \wedge Wife(C, A) \Rightarrow Mother(C, B)$. Let $A = Jack$, $B = Ross$ and $C = Judy$, we get a ground formula: $Father(Jack, Ross) \wedge Wife(Judy, Jack) \Rightarrow Mother(Judy, Ross)$.

*Distance-to-satisfaction.* Denote all ground formulas created by the $q$-th logic rule $F_q$ by $\{G_q^{(j)}\}_{j=1}^{n_q}$, where $n_q$ is the number of ground formula for the $q$-th rule. For any ground formula $G_q^{(j)}$ of $F_q$, when $r_i(h_i, t_i) \in \{0, 1\}$, the satisfaction of $G_q^{(j)}$ can be evaluated via Eq. (1). The value is either 1 or 0, meaning that the ground formula is satisfied or violated, respectively. The binary value of $r_i(h_i, t_i)$ can be relaxed to a continuous value ranging over $[0, 1]$. In such case, we may define the

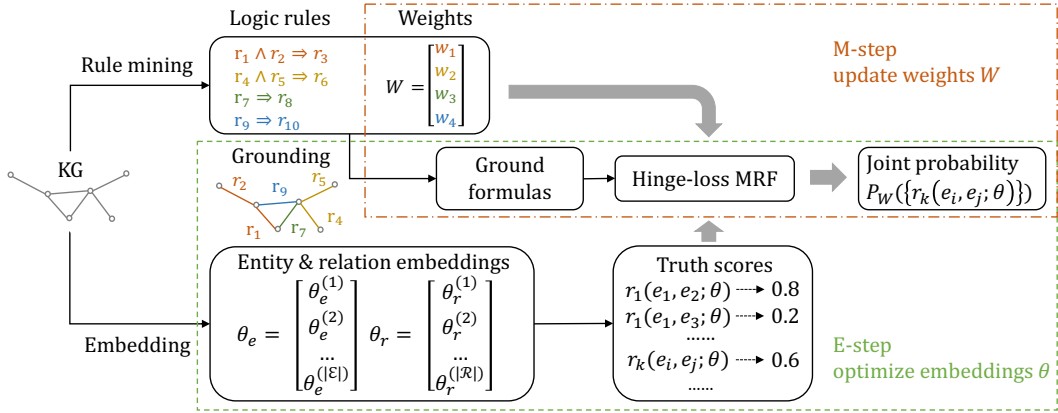

Figure 1: Overall framework of DiffLogic.

*distance-to-satisfaction* for $G_q^{(j)}$ via Łukasiewicz t-norm:

$$d(G_q^{(j)}) := \max\left\{1 - \sum_{i \in \mathcal{I}_q^+} r_i(h_i, t_i) - \sum_{i \in \mathcal{I}_q^-} (1 - r_i(h_i, t_i)),\ 0\right\}. \quad (3)$$

Note that $d(G_q^{(j)}) \in [0, 1]$, and the smaller $d$ is, the better satisfied the ground formula $G_q^{(j)}$ is.

## 3 Differentiable neuro-symbolic reasoning

In this section, we propose a neuro-symbolic model, namely, DiffLogic, which unifies KG-embedding and rule-based reasoning. What follows we present the overall framework of DiffLogic, then show how to perform DiffLogic efficiently.

### 3.1 Overall framework

As shown in Figure 1, the model comprises three components: 1) an efficient grounding technique serving as a filter to identify crucial ground formulas and extract triples connected to them; 2) a KG-embedding model to compute truth scores for the extracted triples; and 3) a tailored continuous MLN framework that takes the truth scores as input and assess the overall probability. The model is optimized using an EM algorithm, alternating between embedding optimization and weight updating. During the E-step, we fix the rule weights and optimize embeddings in an end-to-end fashion, by maximizing the overall probability; while in the M-step, we design an efficient rule weight updating method by leveraging the sparsity of violated rules. It is also worth mentioning here that the model requires a set of rules, which can be obtained from certain rule-mining process or domain experts.

Next, we turn to the abovementioned three components.

**Rule-guided iterative grounding.** The success of probabilistic logic reasoning heavily depends on the grounding process. However, the number of grounding formulas is overwhelmingly large, under our setting, equals $\sum_{q=1}^m |\mathcal{E}|^{|\mathcal{I}_q^-|+1}$. As a result, one has to sample for approximation. In this work, we propose a grounding technique called Rule-guided Iterative Grounding (RGIG) that incrementally identifies crucial ground formulas, reducing the number of ground formulas needed for optimization without compromising performance.

Inference inherently promotes the agreement between assignments (i.e., $\boldsymbol{y}, \boldsymbol{x}$) and the weighted rule set (i.e., $\{(F_q, W_q)\}_{q=1}^m$) by penalizing violated rules. Consequently, ground formulas with higher distance-to-satisfaction are more valuable in guiding optimization. In light of this, we only need to find the ground formulas whose premise atoms in Eq. (2) have (or potentially have) high scores. Pursuing this idea, RGIG iteratively grounds all logic rules and updates a fact set $\mathcal{V}$. The fact set $\mathcal{V}$ is initialized with the observed facts $\mathcal{O}$ from the training set. In each iteration, rules are grounded only from the facts in $\mathcal{V}$ that match the premise parts of the rules. New facts are derived from the conclusion parts of the rules and subsequently added to $\mathcal{V}$. In the subsequent iterations, the updated

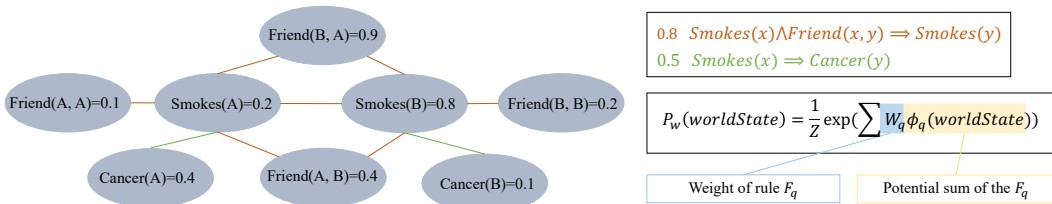

Figure 2: An illustration of probabilistic soft logic (PSL).

$\mathcal{V}$ is used to derive more new facts. From our numerical experience, a few iterations of RGIG are sufficient, say 3. In the end, RGIG yields a set of ground formulas.

This grounding technique leverages the sparsity of violated rules (the distances to satisfaction for most ground formulas are approximately zero) to efficiently identify important ground formulas, facilitating data-efficient optimization. The case study in Section 4.2 illustrates that our grounding technique attains comparable reasoning performance and is orders of $(10^3 \sim 10^5)$ more efficient in terms of the number of ground formulas.

**KG-embedding model.** In the literature of probabilistic logic reasoning, the representation of assignment $\boldsymbol{y}$ for all unobserved facts is very expensive, which is approximate $|\mathcal{E}|^2|\mathcal{R}|$ as the observed fact triples are only a small portion. To this end, we employ a KG-embedding model to parameterize $\boldsymbol{y}$ and $\boldsymbol{x}$. By embedding each entity to continuous representation with $n_e$ parameters, and each relation with $n_r$ parameters. The number of parameters is reduced from $|\mathcal{E}|^2|\mathcal{R}|$ to $|\mathcal{E}|n_e + |\mathcal{R}|n_r$, which is linear with respect to the number of entities/relations.

Although (re)-parameterization improves the representation efficiency, we also need to consider effectiveness. The relation pattern modeling capability is essential for honestly modeling the logic in a KG and performing reasoning. For example, the ability to model symmetric and reverse relation is attributed to capturing logic rules with one premise atom (e.g. $Husband(A, B) \Rightarrow Wife(B, A)$), while the ability to model composition relation is attributed to capturing logic rules with two premise atoms (e.g. $Father(A, B) \land Wife(C, A) \Rightarrow Mother(C, B)$). Among all candidate KG-embedding models, we choose RotatE (Sun et al., 2019) for its simplicity and capability of modeling the various logic patterns. And of course, one may use also other KGE models instead. Let the KGE model be parameterized by $\theta$. Using the sigmoid function, we can transform the score for the fact triple $(h, r, t)$ produced by the KGE model into $[0, 1]$, which can be taken as the truth value for $(h, r, t)$. So the truth value for $(h, r, t)$ can be parameterized by $\theta$. Therefore, the assignment vectors $\boldsymbol{x}$ and $\boldsymbol{y}$ can be parameterized by $\theta$, with their entries being the truth values for the observed and unobserved fact triples, respectively.

**Hinge-loss Markov random field.** Given the parameterized assignments, we employ PSL (Bach et al., 2017) to build a hinge-loss Markov random field (HL-MRF). Specifically, given a knowledge base $\mathcal{K}$ with assignments $\boldsymbol{x}, \boldsymbol{y}$, and a set of weighted logic rules $\{(F_q, W_q)\}_{q=1}^m$. A HL-MRF $P$ over $\boldsymbol{y}$ conditioned on $\boldsymbol{x}$ is a probability density function defined as

$$P_{\boldsymbol{w}}(\boldsymbol{y}|\boldsymbol{x}) = \frac{1}{Z(\boldsymbol{W}, \boldsymbol{x})} \exp(-f_{\boldsymbol{w}}(\boldsymbol{y}, \boldsymbol{x})), \qquad Z(\boldsymbol{W}, \boldsymbol{x}) = \int_{\boldsymbol{y}} \exp(-f_{\boldsymbol{w}}(\boldsymbol{y}, \boldsymbol{x})) \mathrm{d}\boldsymbol{y}, \qquad (4)$$

where $f_{\boldsymbol{w}}$ is the hinge-loss energy function, defined as the weighted sum of all potentials:

$$f_{\boldsymbol{w}}(\boldsymbol{y}, \boldsymbol{x}) = \boldsymbol{W}^\top \boldsymbol{\Phi}(\boldsymbol{y}, \boldsymbol{x}) = [W_1, \dots, W_m][\Phi_1(\boldsymbol{y}, \boldsymbol{x}), \dots, \Phi_m(\boldsymbol{y}, \boldsymbol{x})]^\top = \sum_{q=1}^m W_q \Phi_q(\boldsymbol{y}, \boldsymbol{x}), \quad (5)$$

and $\Phi_q(\boldsymbol{y}, \boldsymbol{x})$ is the sum of potentials of all ground formulas of $F_q$, i.e., $\Phi_q(\boldsymbol{y}, \boldsymbol{x}) = \sum_{j=1}^{n_q} \mathrm{d}(G_q^{(j)})$.

HL-MRF optimizes the assignment $\boldsymbol{y}$ and the rule weights $\boldsymbol{W}$ by alternating between Maximum a posterior (MAP) inference and weight learning. The former step fixes the $\boldsymbol{W}$ and optimizes $\boldsymbol{y}$, while the latter step fixes $\boldsymbol{y}$ and updates $\boldsymbol{W}$.

### 3.2 Optimization

Below we elaborate on the optimization details of DiffLogic. We demonstrate how the model optimization is efficiently performed by employing numerical optimization techniques and approximation methods that leverage sparse properties.

**Embedding updating**    The task in the inference step is to infer $P_{\boldsymbol{w}}(\boldsymbol{y}|\boldsymbol{x})$, i.e., finding the optimum $\boldsymbol{y}$ given the observed assignment $\boldsymbol{x}$ and current weights $\boldsymbol{W}$, which can be formulated as a maximum a posterior (MAP):

$$\mathrm{argmax}_{\boldsymbol{y}}\, P_{\boldsymbol{w}}(\boldsymbol{y}|\boldsymbol{x}) \equiv \mathrm{argmin}_{\boldsymbol{y}\in[0,1]^{|\mathcal{H}|}}\, \boldsymbol{W}^{\top}\boldsymbol{\Phi}(\boldsymbol{y},\boldsymbol{x}).$$

To be consistent with the observation, $\boldsymbol{x}(\theta)$ should be as small as possible. Additionally, we also want the truth values for negative samples to be large. So, the overall cost function can be formulated as:

$$\min_{\theta}\, \boldsymbol{W}^{\top}\boldsymbol{\Phi}(\boldsymbol{y}(\theta),\boldsymbol{x}(\theta)) + \lambda\left(\frac{1}{|T^{+}|}\sum_{(h,r,t)\in T^{+}}[1 - r(h,t;\theta)] + \frac{1}{|T^{-}|}\sum_{(h,r,t)\in T^{-}}r(h,t;\theta)\right),\ (6)$$

where $\boldsymbol{x}$, $\boldsymbol{y}$ are parameterized by $\theta$, $\lambda > 0$ is a parameter, $r(h,t;\theta) \in [0,1]$ stands for the truth value of a fact triple $(h,r,t)$, $T^{+}$ and $T^{-}$ are positive and negative sample sets, respectively. The optimization problem Eq. (6) can be solved by stochastic gradient descent, where we sample a batch of ground formulas to compute the potentials, $T^{+}$ is a batch of positive triples from $\mathcal{O}$, and $T^{-}$ is a set of negative samples obtained by corrupting $T^{+}$.

**Rule weights updating**    In this step, $\theta$ is fixed, and rule weights are updated by maximizing $\log P_{\boldsymbol{w}}(\boldsymbol{y}|\boldsymbol{x})$. The gradient of $\log P_{\boldsymbol{w}}(\boldsymbol{y}|\boldsymbol{x})$ with respect to the weight of $F_{q}$ can be given by

$$\frac{\partial \log P_{\boldsymbol{w}}(\boldsymbol{y}\mid\boldsymbol{x})}{\partial W_{q}} = \mathbb{E}_{\boldsymbol{W}}\left[\Phi_{q}(\boldsymbol{y},\boldsymbol{x})\right] - \Phi_{q}(\boldsymbol{y},\boldsymbol{x}). \tag{7}$$

The first term computes the expectation of potential, which involves integration over $\boldsymbol{y}$ under the distribution defined by rule weight $\boldsymbol{W}$. Directly computing the integration is impractical, so we instead optimize the pseudo-likelihood of training data:

$$P_{\boldsymbol{w}}^{*}(\boldsymbol{y}\mid\boldsymbol{x}) = \prod_{i=1}^{n}\frac{\exp\left[-f_{\boldsymbol{w}}^{i}\left(y_{i}\cup\boldsymbol{y}_{\backslash i},\boldsymbol{x}\right)\right]}{Z_{i}(\boldsymbol{W},y_{i}\cup\boldsymbol{y}_{\backslash i},\boldsymbol{x})}, \quad f_{\boldsymbol{w}}^{i} = \sum_{q=1}^{m}W_{q}\sum_{j=1}^{n_{q}}\mathbf{1}_{\{y_{i}\to G_{q}^{(j)}\}}d(G_{q}^{(j)}).$$

Here $\mathbf{1}_{\{y_{i}\to G_{q}^{(j)}\}} = 1$ if $y_{i}$ is connected to ground formula $G_{q}^{(j)}$, otherwise, zero. $Z_{i}$ is the integration of $f_{\boldsymbol{w}}^{i}$ over $y_{i}$, similar to the second equation in Eq. (4). The pseudo-likelihood is a mean-field approximation for the exact likelihood in Eq. (4). It approximates the exact likelihood by decomposing it into a product of the probabilities of $y_{i}$'s, conditioned on the Markov Blanket of $y_{i}$ denoted by $\mathrm{MB}(y_{i})$. The gradient of the pseudo-likelihood is as follows (see Appendix A.2 for a detailed derivation):

$$\frac{\partial \log P_{\boldsymbol{w}}^{*}(\boldsymbol{y}\mid\boldsymbol{x})}{\partial W_{q}} = \sum_{i=1}^{n}\left\{\mathbb{E}_{y_{i}|\mathrm{MB}}\left[\Psi_{q,MB(i)}\right] - \Psi_{q,MB(i)}\right\}, \ \Psi_{q,MB(i)} = \sum_{j=1}^{n_{q}}\mathbf{1}_{\{y_{i}\to G_{q}^{(j)}\}}d(G_{q}^{(j)}).$$

The above equality enables us to estimate gradients by minibatch sampling. For each sampled $y_{i}$, we need to compute both $\Psi_{q,MB(i)}$ and its expectation. The integration term can be estimated using Monte Carlo integration by fixing other variables and sampling $y_{i}$ on the interval [0, 1]. The Monte Carlo integration is parallelizable and converges quickly as the number of samples increases.

To further reduce the computation burden, we leverage the sparsity of the violated ground formulas to filter ground formulas. Practically, we pre-compute the truth scores of all triples connected to the ground formulas using the KG-embedding model, then use a threshold to divide these triples into a "positive" set and a "negative" set. A ground formula is expected to be violated if all its premise atoms are positive and all its conclusion atoms are negative. By only involving the violated rules when computing potentials, we significantly reduce computational costs.

**Joint reasoning with rules and embeddings**    After training, we get the trained embeddings and updated rule weights, which can be used to perform reasoning. There are two ways to compute the score: 1) compute each embedding score $r_{i}(h_{i},t_{i};\theta)$ using the learned embeddings $\theta$; 2) compute the cumulative rule score $f_{rule}(h_{i},r_{i},r_{i};\boldsymbol{W})$ by summing up weights of ground formulas that infer the target triple:

$$f_{rule}(h_{i},r_{i},t_{i};\boldsymbol{W}) = \sum_{q=1}^{m}W_{q}\sum_{j=1}^{n_{q}}\mathbf{1}_{\{r_{i}(h_{i},t_{i})\text{ can be inferred by }G_{q}^{(j)}\}}.$$

Table 1: Reasoning performance on real-world datasets. [T=n] means the maximum body length of mined rules is n. The best results are shown in bold and the second best are underlined. [NA] indicates that the model cannot finish inference within ten hours.

| | CodeX-s | | CodeX-m | | CodeX-l | | WN18RR | | YAGO3-10 | |
|---|---|---|---|---|---|---|---|---|---|---|
| | MRR | hit@10 | MRR | hit@10 | MRR | hit@10 | MRR | hit@10 | MRR | hit@10 |
| MLP | 0.279 | 0.502 | 0.197 | 0.347 | 0.190 | 0.339 | 0.139 | 0.218 | 0.365 | 0.575 |
| RotatE | 0.421 | 0.634 | 0.325 | 0.466 | 0.319 | 0.453 | 0.469 | 0.566 | 0.495 | 0.670 |
| TuckER | 0.444 | 0.638 | 0.328 | 0.458 | 0.309 | 0.430 | 0.470 | 0.526 | - | - |
| TransE | 0.353 | 0.607 | 0.320 | 0.481 | 0.308 | 0.452 | 0.218 | 0.510 | 0.436 | 0.647 |
| SACN | - | - | - | - | - | - | 0.470 | 0.540 | - | - |
| CompGCN | - | - | - | - | - | - | 0.479 | 0.546 | - | - |
| AMIE | 0.195 | 0.283 | 0.063 | 0.095 | 0.026 | 0.029 | 0.36 | 0.485 | 0.25 | 0.343 |
| NeuraLP | 0.290 | 0.395 | NA | NA | NA | NA | 0.433 | 0.566 | NA | NA |
| DRUM(T=2) | 0.290 | 0.393 | NA | NA | NA | NA | 0.434 | 0.565 | NA | NA |
| DRUM(T=3) | 0.342 | 0.542 | NA | NA | NA | NA | 0.486 | 0.586 | NA | NA |
| RNNLogic$^+$ | - | - | - | - | - | - | 0.51 | 0.597 | NA | NA |
| RLogic$^+$ | - | - | - | - | - | - | **0.52** | **0.604** | **0.53** | **0.703** |
| MLN4KB | 0.082 | 0.134 | 0.035 | 0.045 | 0.028 | 0.032 | 0.368 | 0.374 | 0.460 | 0.525 |
| pLogicNet | 0.342 | 0.505 | 0.306 | 0.448 | 0.270 | 0.388 | 0.440 | 0.534 | 0.387 | 0.595 |
| DiffLogic | 0.445 | 0.662 | 0.335 | 0.487 | 0.326 | 0.448 | 0.493 | 0.585 | 0.503 | 0.673 |
| DiffLogic$^+$ | **0.458** | **0.655** | **0.343** | **0.495** | **0.337** | **0.46** | 0.50 | 0.587 | 0.513 | 0.674 |

To perform joint inference using both embedding scores and rule scores, a simple way is to use the weighted sum of embedding scores and the normalized rule scores:

$$r_i(h_i, t_i) = (1 - \eta) \cdot r_i(h_i, t_i; \theta) + \eta \cdot \widehat{f}_i(h_i, t_i; \boldsymbol{W}), \qquad (8)$$

where $\widehat{f}_i(h_i, t_i; \boldsymbol{W})$ is the rescaled value of $f_{rule}(h_i, r_i, t_i; \boldsymbol{W})$ calculated by minmax normalization. The optimum weight coefficient $\eta$ is selected by using the validation set.

# 4 Experiments

In this section, we conduct experiments to answer the following research questions. **RQ1:** Can DiffLogic outperform rule-based and embedding-based models in terms of reasoning performance? **RQ2:** Can DiffLogic scale to large knowledge graphs that rule-based models struggle to handle? **RQ3:** Does DiffLogic actually learn representations compatible with rules? **RQ4:** Quantify the efficiency and scalability of DiffLogic. **RQ5:** How effective is the model in terms of leveraging prior knowledge encoded in rules, compared with data-driven methods?

**Datasets and candidate rules.** We incorporate four real-world knowledge graph datasets: YAGO3-10, WN18, WN18RR, and CodeX (available in three sizes: small, medium, and large), along with a synthetic logic reasoning dataset: Kinship. Dataset statistics and descriptions can be found in Appendix B.1. Candidate rules for knowledge graphs are mined using AMIE3 (Lajus et al., 2020), with rule weights initialized by rule confidence scores.

**Baseline models.** Baseline models include four KG-embedding models — TransE (Bordes et al., 2013), RotatE (Sun et al., 2019), TuckER (Balažević et al., 2019), MLP (Dong et al., 2014), two GNN-based models — SACN (Shang et al., 2019), CompGCN (Vashishth et al., 2019), four rule-learning models — Neural LP (Yang et al., 2017), DRUM (Sadeghian et al., 2019), RNNLogic (Qu et al., 2020), RLogic (Cheng et al., 2022), a discrete MLN engine MLN4KB Fang et al. (2023), and a neuro-symbolic model pLogicNet (Qu & Tang, 2019) that also integrate KG-embedding and MLN. We exclude the recently proposed ExpressGNN (Zhang et al., 2020) from our experiments on KG experiments since it requires querying test data[1] during training and is inapplicable in our setting. For all baselines that employ a KG embedding model, we unify the negative sampling scheme as adversarial negative sampling for fair comparison. Hyperparameters for each baseline are taken from their original paper.

---

[1]Please see the discussion on the usage of test data at `https://openreview.net/forum?id=rJg76kStwH`.

## 4.1 Reasoning on real-world knowledge graphs

To answer **RQ1**, **RQ2**, and **RQ3**, we conduct link prediction tasks on several real-world datasets. Including WN18RR, YAGO3-10, and CodeX of three sizes (denoted as CodeX-s/m/l). Note that YAGO3-10 and Codex-l are large knowledge graphs that are suitable for assessing scalability. We report the performance of DiffLogic under two settings, using embedding scores for reasoning (denoted by DiffLogic) and using both embedding scores and rule scores for reasoning (denoted by DiffLogic$^+$). We evaluate the performance using the Mean Reciprocal Rank (MRR) and Hit@10, with the results presented in Table 1. We also investigate the evolution of violated rules and MRR on WN18RR during the inference of DiffLogic and RotatE, as demonstrated in Figure 3. These results lead to several key observations:

**First, DiffLogic surpasses both rule-based and KG embedding-based methods.** This can be primarily attributed to the fact that it combines the ability of both sides: 1) the ability to utilize embeddings to model similarities among entities, which enhance the reasoning performance; and 2) compared with data-driven KG-embedding models that only learn rule patterns from a large amount of labeled data, DiffLogic can explicitly leverage rule patterns within a principled logic reasoning framework, leading to better overall performance. Our results also indicate that jointly using rules and learned embeddings for reasoning is more effective than solely relying on embeddings. This suggests that rules and embeddings can complement each other during the reasoning process, ultimately leading to more accurate and robust inferences.

**Second, DiffLogic outperforms pLogicNet, a neuro-symbolic model.** Although pLogicNet also integrates MLN and KG-embeddings for reasoning, the key difference is that DiffLogic optimizes MLN and KG-embedding using a unified objective, whereas pLogicNet accommodates the discrete nature of MLN and essentially employs an MLN as a data augmentation technique to annotate additional facts for training KG embeddings. As a result, the optimization of MLN and KG embedding in pLogicNet is less consistent, and its performance is sensitive to the annotation threshold. Its performance is lower than TransE - its base KG embedding model on several large KGs (YAGO3-10/CodeX-l) because the annotated triples are mostly false positive. This highlights the advantages of DiffLogic in providing a more coherent and robust optimization process.

**Third, DiffLogic demonstrates superior scalability compared to rule-based methods.** DiffLogic is adept at efficiently scaling to expansive KG datasets like YAGO3-10 and CodeX-l. Conversely, rule-based methods, including NeuraLP, DRUM, RNNLogic, and MLN4KB, often encounter challenges when attempting to scale to such large KGs. An exception is RLogic, which is a scalable rule-learning model designed to mine complex and lengthy rules for reasoning. Remarkably, our model achieves comparable results by including only simply rules with a rule body length of $\leq 2$.

**Furthermore, DiffLogic proficiently learns representations that align with both KG-embeddings and rules.** As the left subfigure in Figure 3 demonstrates, the number of violated rules during the inference (embedding learning) phase is initially sparse, rises rapidly, and eventually decreases with training progression. We attribute this to 1) the random initialization of embeddings, which assigns low truth scores to most triples, resulting in fewer initial violations; 2) the model's training phase, where truth scores for training set triples increase but rule patterns are not yet fully captured, causing a rapid rise in rule violations; 3) as training progresses, KG embeddings begin to encapsulate rule patterns, reducing the number of violations. Importantly, despite its base embedding model being RotatE, DiffLogic is more effective than pure data-driven RotatE in capturing rule patterns, ensuring a consistent decrease in rule violations. As depicted in the middle subfigure in Figure 3, DiffLogic achieves faster convergence in test MRR compared to RotatE.

## 4.2 Scalability of optimization

To answer **RQ4**, we assess the scalability of our methodology, focusing primarily on optimization efficiency, i.e., the efficacy of grounding. We employ Kinship, a widely used (Zhang et al., 2020; Fang et al., 2023) synthetic benchmark dataset, to evaluate the efficacy and efficiency of RGIG in identifying important ground formulas. This dataset includes a training set, a test set, and a set of logic rules with full confidence. Some predicates are unobserved in the training set and can only be inferred via logic rules. The task is to deduce the gender of each individual in the test set, given the training set and the rule set. Kinship, designed for logical reasoning, necessitates resolving contradictions

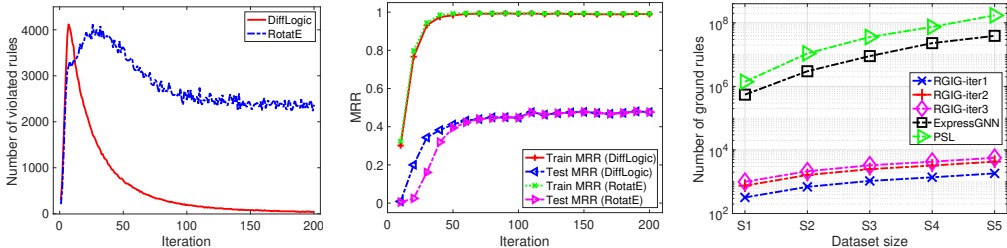

Figure 3: *Left*: Violated rules evolution during inference on WN18RR. *Middle*: MRR evolution on WN18RR. *Right*: Number of considered ground formulas in PSL/ExpressGNN and RGIG.

among ground formulas, thus the reasoning performance heavily depends on the grounding process. Such a challenging dataset is suitable for assessing the efficacy of grounding techniques.

In our empirical evaluations, we discovered that three iterations of RGIG are enough to identify crucial ground formulas for accurate reasoning, yielding an AUC-ROC of $0.982 \pm 0.014$ (detailed results are available in Appendix B.2.). We then compared the efficiency of RGIG with PSL and ExpressGNN's grounding techniques by examining the number of ground formulas considered for optimization across five different sizes of the Kinship dataset. Detailed calculations for PSL grounding can be found in Appendix A.3. The comparative results are depicted in the right subfigure in Figure 3. Although PSL and ExpressGNN can also perform accurate reasoning on Kinship, our model using RGIG has orders of $(10^3 \sim 10^5)$ more data-efficient optimization process over varied sizes of Kinship dataset, demonstrating its applicability.

To further evaluate the efficiency of RGIG, we empirically test the run-time and memory overhead on real-world datasets. Results are presented in Table 2.

Table 2: Grounding overhead on real-world datasets. Run-time overhead is evaluated ten times and report mean and std.

| Datasets | CodeX-s | CodeX-m | CodeX-l | WN18RR | YAGO3-10 |
|---|---|---|---|---|---|
| Run-time(/sec) | 0.03±0.00 | 0.38±0.01 | 0.87±0.04 | 0.54±0.01 | 3.20±0.04 |
| Memory(/MB) | 2.19 | 11.57 | 25.58 | 18.73 | 262.65 |

The experimental results demonstrate that RGIG can efficiently scale to accommodate large KGs while maintaining minimal run-time and memory overhead. On the largest KG, YAGO3-10, the grounding process is completed in approximately 3.2 seconds, using around 262 MB of memory. In practice, we also observed that as the grounding iteration progresses, most inferred facts turn out to be negative triples. This trend could introduce noise and potentially impact efficiency. To enhance the efficiency of RGIG, one could consider filtering out facts with low scores by using a pre-trained RotatE model.

### 4.3 Learning from data vs. learning from rules

To answer **RQ5**, we design a "rule-pattern re-injection" experiment to evaluate DiffLogic's capability of injecting prior knowledge into embeddings, and compare it with pure data-driven based KG-embeddings. The design details are as follows:

The experiment contains two steps: 1) rule pattern removal, and 2) rule pattern re-injection. In our experiments, we use WN18 and select fourteen rules whose confidence scores are higher than 0.95. Then we deduplicate these rules so that no rules can be inferred from other rules, resulting in seven compact rules. In the **rule pattern removal step**, we use the selected seven rules to split the original training set, by finding all paths in the original training set that match the rules and extract the connected triples. The triples that match the conclusion part of the rules comprise the pattern set, and the original training set with the pattern set removed becomes the fact set. In this way, the generated fact set does not contain any pattern for the seven rules. In the **rule pattern re-injection step**, we add different ratios ($0\%, 10\%, 20\%, 100\%$) of the pattern set back into the fact set, so that rule patterns become increasingly evident. For DiffLogic, seven rules are applied to the training process for explicit rule pattern injection. We also include three KG-embedding models

for comparison. To fairly compare the rule-injection capability for embeddings, we use only the embedding scores of DiffLogic for evaluation. Experimental results are presented in Table 3.

Table 3: Comparison of pure data-driven training and rule injection by DiffLogic for embeddings.

| Model | MRR | | | | Hits@10 | | | |
|---|---|---|---|---|---|---|---|---|
| | 0% | 10% | 20% | 100% | 0% | 10% | 20% | 100% |
| MLP | 0.123 | 0.156 | 0.198 | 0.851 | 0.279 | 0.364 | 0.449 | 0.932 |
| TransE | 0.399 | 0.469 | 0.500 | 0.775 | 0.917 | 0.944 | 0.944 | 0.957 |
| RotatE | 0.579 | 0.890 | 0.927 | 0.944 | 0.784 | 0.949 | 0.961 | 0.962 |
| DiffLogic | **0.954** | **0.953** | **0.952** | **0.954** | **0.964** | **0.966** | **0.963** | **0.967** |

The results show a significant difference between the two rule pattern learning paradigms: DiffLogic can directly leverage explicit prior knowledge compactly encoded in rules, achieving significant improvement in reasoning performance by using only a small number of logical rules. On the contrary, pure data-driven based KG-embedding algorithms can only implicitly learn rule patterns from labeled data. By adding more data from the pattern set back to the fact set, the performance of data-driven algorithms increases as the rule patterns become more evident in the knowledge base. Nevertheless, these data-driven algorithms are not comparable to DiffLogic even though rule patterns are observable.

## 5    Related work

There have been some studies attempting to integrate rule-based methods and KG-embedding models. For example, Guo et al. (2016) proposes to learn embeddings from both triples and rules by treating triples as 'atomic formulas' while treating ground logic formulas as 'complex formulas', thus unifying the learning from triples and rules. However, their framework only uses hard rules and thus cannot make use of the soft rules with uncertainty. Another study (Guo et al., 2018) applied soft rules for generating additional training data but could not optimize rule weights, making the model's effectiveness dependent on rule weights initialization. Qu & Tang (2019) enable embedding learning and rule weight updating in a tailored MLN framework, by alternately employing one component to annotate triples to update the other component. However, the annotation process renders the inference not differentiable and is sensitive to the annotation threshold. Moreover, their grounding process only considers ground formulas with premise atoms observed in the training set, limiting the model's effectiveness due to the potential omission of important ground formulas. Zhang et al. (2020) designed a graph neural network for learning structure-aware expressive representations under the MLN framework. However, its inference efficiency suffers from a large number of ground formulas and limited generalization ability (requiring querying the test data during inference). The MLN-based neuro-symbolic models (Qu & Tang, 2019; Zhang et al., 2020) benefit from the dynamically updated rule weights which handle uncertainty. However, they all fail to directly optimize the objective of MLN due to the complexity of the associated integration and thereby resorting to optimizing ELBO.

## 6    Conclusion

In this paper, we aim to scale neuro-symbolic reasoning to large knowledge graphs with improved performance. We develop a differentiable model, namely, DiffLogic, that combines the advantages of KG-embedding models and rule-based models. DiffLogic directly optimizes the joint probability rather than the EBLO. The KG-embedding component enables linear scalability in representation, together with the grounding technique and the estimation technique for the gradient of rule weights, make DiffLogic efficient and scalable. Numerical simulations on benchmark datasets show the merits of DiffLogic. The performance of DiffLogic heavily depends on the quality of the rules. Was it possible to design an automatic and differentiable rule mining method, we may incorporate it into DiffLogic to further improve the performance. This could be future work.

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

# A  Notations and mathematical proofs

## A.1  Notations

Table 4: Notations.

| Notation | Description |
|---|---|
| $\mathcal{K}$ | The knowledge base |
| $\mathcal{E}$ | The entity set |
| $\mathcal{R}$ | The relation set |
| $\mathcal{O}, \mathcal{H}$ | The set of observed and unobserved facts |
| $\boldsymbol{x}, \boldsymbol{y}$ | The assignments of $\mathcal{O}$ and $\mathcal{H}$, respectively |
| $\{F_q, W_q\}_{q=1}^m$ | The set of logic rules and attached weights |
| $\mathcal{I}_q^-, \mathcal{I}_q^+$ | The index set of premise atoms and conclusion atoms of rule $F_q$, respectively |
| $A, B, ...$ | Variables in logic rules |
| $\{G_q^{(j)}, j \in t_q\}$ | All ground formulas created by the $q_{th}$ logic rule |
| $\Phi_q(\boldsymbol{y}, \boldsymbol{x})$ | The sum of potentials of all ground formulas of $F_q$ |
| $\theta$ | The embedding parameters |

## A.2  Derivation of rule weight gradient

Given

$$P_{\boldsymbol{w}}^*(\boldsymbol{y} \mid \boldsymbol{x}) = \prod_{i=1}^n P^*(y_i \mid \mathrm{MB}(y_i), \boldsymbol{x}) = \prod_{i=1}^n \frac{\exp\left[-f_{\boldsymbol{w}}^i\left(y_i \cup \boldsymbol{y}_{\backslash i}, \boldsymbol{x}\right)\right]}{Z_i(\boldsymbol{W}, y_i \cup \boldsymbol{y}_{\backslash i}, \boldsymbol{x})},$$

$$Z_i(\boldsymbol{W}, y_i \cup \boldsymbol{y}_{\backslash i}, \boldsymbol{x}) = \int_{y_i} \exp\left[-f_{\boldsymbol{w}}^i\left(y_i \cup \boldsymbol{y}_{\backslash i}, \boldsymbol{x}\right)\right], \quad f_{\boldsymbol{w}}^i = \sum_{q=1}^m W_q \sum_{j=1}^{n_q} \mathbf{1}_{\{y_i \to G_q^{(j)}\}} d(G_q^{(j)}),$$

we have

$$\frac{\partial \log P^*(\boldsymbol{y} \mid \boldsymbol{x})}{\partial W_q} = \sum_{i=1}^n \frac{\partial \log P^*(y_i \mid \mathrm{MB}(y_i), \boldsymbol{x})}{\partial W_q}. \tag{9}$$

The partial derivative in the left side of Eq. (9) is a summation of $n$ terms, each term represents the partial derivatives of the pseudo-log-likelihood for each $y_i$, conditioned on its Markov blankets. Each term can be further simplified as follows:

$$\frac{\partial \log P^*(y_i \mid \mathrm{MB}(y_i), \boldsymbol{x})}{\partial W_q}$$

$$= \frac{\partial \left\{-f_{\boldsymbol{w}}^i\left(y_i \cup \boldsymbol{y}_{\backslash i}, \boldsymbol{x}\right) - \log Z_i(\boldsymbol{W}, y_i \cup \boldsymbol{y}_{\backslash i}, \boldsymbol{x})\right\}}{\partial W_q}$$

$$= \frac{\partial \left\{-f_{\boldsymbol{w}}^i\left(y_i \cup \boldsymbol{y}_{\backslash i}, \boldsymbol{x}\right) - \log \int_{y_i} \exp\left[-f_{\boldsymbol{w}}^i\left(y_i \cup \boldsymbol{y}_{\backslash i}, \boldsymbol{x}\right)\right]\right\}}{\partial W_q}.$$

Here, we can easily get

$$\frac{\partial f_{\boldsymbol{w}}^i\left(y_i \cup \boldsymbol{y}_{\backslash i}, \boldsymbol{x}\right)}{\partial W_q} = \sum_j \mathbf{1}_{\{y_i \to G_q^{(j)}\}} d(G_q^{(j)}). \tag{10}$$

To make the writing concise, we replace the right term of Eq. (10) with the following notation:

$$\Psi_{q, MB(i)} = \sum_j \mathbf{1}_{\{y_i \to G_q^{(j)}\}} d(G_q^{(j)}).$$

In this way, we can deduce that:

$$\frac{\partial \log P^*(y_i \mid \mathrm{MB}\,(y_i)\,, \boldsymbol{x})}{\partial W_q}$$
$$= - \Psi_{q,MB(i)} - \frac{1}{Z_i(\boldsymbol{W}, y_i \cup \boldsymbol{y}_{\backslash i}, \boldsymbol{x})} \frac{\partial \int_{y_i} \exp\left[-f_{\boldsymbol{w}}^i\left(y_i \cup \boldsymbol{y}_{\backslash i}, \boldsymbol{x}\right)\right]}{\partial W_q}. \tag{11}$$

The partial derivative and the integration in Eq. (11) can be swapped using Lebesgue's dominated convergence theorem, the Eq. (11) thus becomes:

$$\frac{\partial \log P^*(y_i \mid \mathrm{MB}\,(y_i)\,, \boldsymbol{x})}{\partial W_q}$$
$$= - \Psi_{q,MB(i)} - \frac{1}{Z_i(\boldsymbol{W}, y_i \cup \boldsymbol{y}_{\backslash i}, \boldsymbol{x})} \int_{y_i} \frac{\partial \exp\left[-f_{\boldsymbol{w}}^i\left(y_i \cup \boldsymbol{y}_{\backslash i}, \boldsymbol{x}\right)\right]}{\partial W_q}$$
$$= - \Psi_{q,MB(i)} + \int_{y_i} \frac{\exp\left[-f_{\boldsymbol{w}}^i\left(y_i \cup \boldsymbol{y}_{\backslash i}, \boldsymbol{x}\right)\right]}{Z_i(\boldsymbol{W}, y_i \cup \boldsymbol{y}_{\backslash i}, \boldsymbol{x})} \Psi_{q,MB(i)}$$
$$= - \Psi_{q,MB(i)} + \int_{y_i} P^*\left(y_i \mid \mathrm{MB}\,(y_i)\,, \boldsymbol{x}\right) \Psi_{q,MB(i)}$$
$$= - \Psi_{q,MB(i)} + \mathbb{E}_{y_i \mid \mathrm{MB}}\left[\Psi_{q,MB(i)}\right].$$

Therefore, the partial derivative of pseudo-log-likelihood with respect to rule weight $W_q$ is computed by:

$$\frac{\partial \log P^*(\boldsymbol{y} \mid \boldsymbol{x})}{\partial W_q} = \sum_{i=1}^{n} \left\{ \mathbb{E}_{y_i \mid \mathrm{MB}} \left[ \sum_j \mathbf{1}_{\{y_i \to G_q^{(j)}\}} d(G_q^{(j)}) \right] - \sum_j \mathbf{1}_{\{y_i \to G_q^{(j)}\}} d(G_q^{(j)}) \right\}.$$

### A.3 Calculation of number of ground formulas for Kinship datasets

We present a detailed calculation of the number of ground formulas considered by PSL in Kinship datasets as follows.

Given

- a first-order logical rule $F_q$ containing $|\mathcal{I}_q^-|$ premise atoms, and
- a knowledge base containing $|\mathcal{E}|$ number of entities,

the number of variables in $F_q$ is $|\mathcal{I}_q^-| + 1$.

PSL grounds each rule by substituting the variables with all possible entities. The number of ground formulas created by this logic rule $F_q$ on the knowledge base is:

$$|\mathcal{E}|^{|\mathcal{I}_q^-|+1}.$$

Thus the overall ground formulas created by the rule set $\{F_q\}_{q=1}^m$ is:

$$\sum_{q=1}^{m} |\mathcal{E}|^{|\mathcal{I}_q^-|+1}.$$

Given the statistics of Kinship datasets in Table 7, rules statistics are shared across different sizes of Kinship datasets, each dataset contains 12 rules that contain two variables and 9 rules that contain 3 variables. The number of ground formulas considered by PSL is thus computed by:

$$12 \times |\mathcal{E}|^2 + 9 \times |\mathcal{E}|^3. \tag{12}$$

By applying the Eq. (12), we can get the ground formula number for each size of the Kinship dataset, as presented in Table 5:

Table 5: Number of ground formulas of Kinship datasets created by classical grounding method.

| Kinship Size | S1 | S2 | S3 | S4 | S5 |
|---|---|---|---|---|---|
| Number of ground formulas | 1,373,601 | 10,853,976 | 35,798,376 | 74,671,320 | 172,162,935 |

# B    Experimental details

## B.1    Dataset statistics

We list the statistics of the real-world knowledge graph datasets in Table 6 and the synthetic Kinship dataset in Table 7. We present detailed descriptions for each dataset below.

*CodeX.* The CodeX dataset, recently proposed for knowledge graph completion tasks, is a comprehensive collection extracted from both Wikidata and Wikipedia. This challenging dataset comes in three versions: small (S), medium (M), and large (L), allowing for comprehensive evaluation.

*YAGO3-10.* YAGO3-10 is a subset of YAGO3 (Suchanek et al., 2007), a large knowledge base completion dataset, with the majority of triples describing attributes of persons, including their citizenship, gender, and profession.

*WN18.* WordNet 18 (WN18) dataset is one of the most commonly used subsets of WordNet.

*WN18RR.* WN18RR is a modified version of WN18 designed to be more challenging for knowledge graph reasoning algorithms by removing reverse relations in the knowledge graph.

*Kinship.* A synthetic dataset, widely used (Zhang et al., 2020; Fang et al., 2023) for evaluating the statistical relational learning ability and the scalability of reasoning algorithms. We use five different sizes of the dataset for evaluating its run time efficiency and parameter scalability, namely Kinship-S1/S2/S3/S4/S5, respectively.

Table 6: Statistics of real-world knowledge base datasets.

| Dataset | #Ent | #Rel | #Train/Valid/Test | #Rules |
|---|---|---|---|---|
| CodeX-s | 2,034 | 42 | 32,888/1,827/1,828 | 35 |
| CodeX-m | 17,050 | 51 | 185,584/10,310/10,311 | 52 |
| CodeX-l | 77,951 | 69 | 551,193/30,622/30,622 | 57 |
| YAGO3-10 | 123,182 | 37 | 1,079,040/5,000/5,000 | 22 |
| WN18 | 40,943 | 18 | 141,442/ 5,000/ 5,000 | 140 |
| WN18RR | 40,943 | 11 | 86,835/ 3,034/ 3,134 | 51 |

Table 7: Statistics for Kinship datasets of varied sizes (S1-S5).

| | S1 | S2 | S3 | S4 | S5 |
|---|---|---|---|---|---|
| Number of rules containing 1 premise atom | 12 | 12 | 12 | 12 | 12 |
| Number of rules containing 2 premise atoms | 9 | 9 | 9 | 9 | 9 |
| Number of predicates | 15 | 15 | 15 | 15 | 15 |
| Number of entities | 52 | 106 | 158 | 202 | 267 |

## B.2    Probabilistic logic reasoning on Kinship Dataset

We assess performance on the Kinship dataset across five different sizes. Due to the full confidence of rules, we only perform inference in this experiment and do not need to update weights. We include

Table 8: Comparative evaluation of reasoning performance on the Kinship dataset.

| Algorithms | Ground iteration | AUC-ROC | | | | |
|---|---|---|---|---|---|---|
| | | S1 | S2 | S3 | S4 | S5 |
| PSL | - | .976±.011 | .980±.005 | .991±.003 | .982±.005 | .972±.004 |
| ExpressGNN | - | .957±.002 | .921±.001 | .959±.004 | .940±.001 | .989±.004 |
| DiffLogic-RotatE | 1 | .841±.005 | .895±.001 | .922±.001 | .901±.001 | .903±.000 |
| | 2 | .931±.005 | .994±.001 | .998±.001 | .985±.001 | .993±.001 |
| | 3 | .937±.005 | .987±.001 | .995±.001 | .978±.001 | .989±.001 |
| DiffLogic-MLP | 1 | .567±.099 | .537±.041 | .507±.024 | .503±.018 | .504±.014 |
| | 2 | .956±.032 | .997±.002 | .999±.003 | .999±.001 | **.999±.000** |
| | 3 | **.982±.014** | **.997±.001** | **.999±.001** | **.999±.000** | **.999±.000** |

DiffLogic using two different embedding models, i.e., RotatE and MLP, and evaluate their reasoning performance using RGIG with varied iterations (i.e., 1, 2, 3) for grounding. We include PSL and ExpressGNN as baselines, but we exclude pLogicNet due to its inability to utilize handcrafted rules. Given that the Kinship dataset lacks a validation set, we run each model ten times and report the AUC-ROC statistics from the final epoch of each run on the test set. The results are presented in Table 8, with the best results shown in bold.

## B.3 Comparing inference time on Kinship

We evaluate the inference time on the Kinship dataset across five different sizes. We include models in Appendix B.2 for this experiment. For two DiffLogic variants, we only evaluate their inference time when using 3 iterations of RGIG for grounding. All the runtime experiments are conducted in the same machine with configurations as in Table 9. All of these models are implemented in Python, thereby ensuring a fair comparison. The inference time results are displayed in Table 10, with the best results shown in bold.

Table 9: Machine configuration.

| Component | Specification |
|---|---|
| GPU | NVIDIA GeForce RTX 3090 |
| CPU | Intel(R) Xeon(R) Silver 4214R CPU @ 2.40GHz |

Table 10: Comparison of runtime of inference on Kinship.

| Algorithms | Grounding iteration | Runtime | | | | |
|---|---|---|---|---|---|---|
| | | S1 | S2 | S3 | S4 | S5 |
| PSL | - | ~3.6min | ~7.9min | ~12.9min | ~13.5min | ~32min |
| ExpressGNN | - | ~18.4min | ~19.1min | ~18.9min | ~19.4min | ~20.2min |
| DiffLogic-RotatE | 3 | 37s | ~1.5min | ~3.2min | ~3.6min | ~4min |
| DiffLogic-MLP | 3 | **21.8s** | **41.5s** | **45s** | **54.4s** | **~1.2min** |

