# OpenReview forum: "Differentiable Neuro-Symbolic Reasoning on Large-Scale Knowledge Graphs"
_NeurIPS.cc/2023/Conference — NeurIPS 2023 poster_

### Official Review · Reviewer_wvKd · 2023-06-27

**Soundness:** 3 good
**Presentation:** 3 good
**Contribution:** 3 good
**Rating:** 7
**Confidence:** 4

**Summary:**

This paper integrates rule-based reasoning and knowledge graph (KG) embedding to enable effective and efficient knowledge graph reasoning. The key idea is to use probabilistic soft logic (PSL) to assess the agreement between the inferred triples and weighted rules, based on the embedding representations of entities and relations. The proposed framework DiffLogic uses several mechanisms to accelerate the optimization. First, it selects essential triples from the inferred triples to perform the assessment. Second, it utilizes the sparsity of violated triples to efficiently estimate the gradient of rule weights. Extensive experiments have been conducted, and the results demonstrate that DiffLogic outperforms all state-of-the-art baselines.

**Strengths:**

- Originality: It is novel to use probabilistic soft logic (PSL) to perform neuro-symbolic reasoning, which makes the proposed framework DiffLogic differentiable. Two tailored mechanisms have been designed to accelerate the optimization. First, it selects essential triples from the inferred triples to perform the assessment. Second, it utilizes the sparsity of violated triples to efficiently estimate the gradient of rule weights.

- Quality: The proposed framework is technically sound. Extensive experiments have been conducted, and the results demonstrate that DiffLogic outperforms state-of-the-art baselines. Details of the experiments have been provided.

- Clarity: The motivation and challenges are well described. The main idea of the proposed framework is well explained. The Experiments Section is well structured.

- Significance: Neuro-symbolic reasoning is promising since it can potentially combine the advantages of KG embedding and rule-based reasoning. The experimental results show that DiffLogic is effective and efficient.

**Weaknesses:**

1. Some notations are confusing. First, examples should be given to explain I^-_q and I^+_q. Why distinguishing them are important to DiffLogic? Second, we already have h and t to represent head and tail. Why do we still need A and B? Third, it seems like Z() is unnecessary since it is only used in Eq. (4). Fourth, \theta denotes the embedding parameters. Then, what are the meanings of x(\theta) and y(\theta)? Should we use x|\theta and y|\theta?

2. The writing can be improved. First, examples should be given to explain assignments x and y. Are they continuous? Second, it seems like ground formulas are triples, but the name “formula” is confusing. Third, the connection between contribution (1) and contributions (2) (3) (4) is unclear. What is ELBO, and why is it related to DiffLogic? What are the relations between DiffLogic and “efficient grounding technique”?

3. The paper emphasizes that DiffLogic is efficient. The running time results in Table 9 should be moved from Supplementary Material to Main Content. The time complexity of DiffLogic should be given?

4. All equations should be indexed.

**Questions:**

1. Why distinguishing I^-_q and I^+_q are important to DiffLogic?

2. What are the meanings of x(\theta) and y(\theta)? Should we use x|\theta and y|\theta?

3. What is ELBO, and why is it related to DiffLogic?

4. What is the corresponding mathematical expression for using important ground formulas to perform the assessment?

5. What is the time complexity of DiffLogic in Big O?


==
I acknowledged I have read the rebuttal and decide to keep my score.

**Limitations:**

The authors have already mentioned the limitation, i.e., “The performance of DiffLogic heavily depends on the quality of the rules”. It is the limitation of all rule-based reasoning studies and neuro-symbolic reasoning studies.

---

> ### Author Rebuttal · Authors · 2023-08-09
>
> Q1. Why distinguishing $I^-_q$ and $I^+_q$ are important to DiffLogic?
>
> A.
> Thank you for your question.
>
> The importance of differentiating between the notations $I^-_q$ and $I^+_q$ is twofold: knowledge representation and logical reasoning.
>
> 1. Knowledge Representation: As introduced in subsection 2.2 of our paper, a first-order logic is represented as a disjunction of atoms and their negations. $I^-_q$ and $I^+_q$ are index sets containing the indices of atoms that are negated or not, respectively. A rule in this format can also be reorganized as an implication from the premise (negated atoms) to the conclusion (non-negated atoms). Distinguishing between these two notations is crucial as they designate which part forms the premises and which part forms the conclusion.
>
> 2. Logical Reasoning: When employing a logic rule for reasoning, it's intrinsic to identify facts that match the premise. Once all facts from the knowledge graph match the premise, the facts in the conclusion part are then inferred.
>
> DiffLogic jointly uses a set of weighted rules to perform reasoning on knowledge graphs. During rule grounding, we find paths that match the negated atoms of a rule. During probabilistic logic reasoning, the assignments y and x are encouraged to satisfy more rules so more new facts are inferred from the knowledge graph.
>
> Q2. What are the meanings of $x(\theta)$ and $y(\theta)$? Should we use $x_\theta$ and $y_\theta$?
>
> A.
> Thanks for your question.
>
> The notations $x(\theta)$ and $y(\theta)$ represent the assignments $x$ and $y$, both parameterized by an embedding $\theta$. In the context of Probabilistic Soft Logic (PSL), the inference task is to infer the assignment of unobserved facts, or $y$, based on the assignments of observed facts, $x$.
>
> Under the MLN/PSL framework, the full representation of all assignments is memory-intensive — it requires $O(|\mathcal{E}|^2|\mathcal{R}|)$ parameters to represent all assignments for a Knowledge Graph (KG) with $\mathcal{E}$ entities and $\mathcal{R}$ relations—we optimize this by parameterizing these assignments through the output scores of a KG embedding model. As such, the assignments become $x(\theta)$ and $y(\theta)$.
>
> During the inference step, we update the embedding $\theta$, so it's necessary to explicitly represent all assignments as $x(\theta)$ and $y(\theta)$. Conversely, during the weight updating step, the embeddings $\theta$ are fixed. Therefore, we can omit it for concise writing.
>
> I hope this clarifies your query about our notation.
>
> Q3. What is ELBO, and why is it related to DiffLogic?
>
> A.
> In variational inference, the Evidence Lower BOund (ELBO) is a crucial concept. It's a function of the parameters of the variational distribution.
>
> The idea behind variational inference is to approximate the true posterior distribution (which is often intractable in complex models) with a simpler distribution that we can work with more easily. The process aims to minimize the Kullback-Leibler (KL) divergence between the approximated and the true posterior, or alternatively, to maximize the ELBO.
>
> In terms of existing Markov Logic Network (MLN)-based neuro-symbolic methods, such as pLogicNet and ExpressGNN, they employ either a KG embedding model or a Graph Neural Network (GNN) to approximate the optimal distribution of assignments x and y. The optimization of this approximation occurs through updating embeddings. Therefore, the embedding updating step is actually a variational inference. Both pLogicNet and ExpressGNN seek to optimize ELBO as their objective to avoid intractable computation, but this also leads to in-direct optimization of the actual objective of MLN.
>
> Contrastingly, DiffLogic allows us to directly maximize the posterior (in equation (4)) due to the continuous characteristic of our framework, leading to a more smooth integration of MLN and KG embedding model, and better optimization efficiency.
>
> Q4. What is the corresponding mathematical expression for using important ground formulas to perform the assessment?
>
> A.
> Thank you for your question.
>
> In this work, we use rule-guided iterative grounding to identify important ground formulas, to facilitate efficient optimization, rather than assessment.
>
> Using the important ground formulas for optimization is shown in equation (6). The first term is computing the weighted sum of potential, which is computed on the selected ground formulas.
>
> Q5. What is the time complexity of DiffLogic in Big O?
>
> A.
> Thanks for your question.
>
> The run-time of DiffLogic consists of two parts, inference (embedding updating) and rule weight updating:
>
> The Big O time complexity for inference is  $O(n*e/bs)$, where $n$ is the number of triples in the training set, $e$ is the overall number of epochs of embedding learning, and the $bs$ is the number of training data used in each batch.
>
> The Big O time complexity for rule weight updating is $O(b_w\*n_m)$, here $b_w$ denotes the batch size we use to estimate the gradient weights (see the equation between line 209 and line 210) in minibatch, and $n_m$ is the sample size we use in Monte Carlo integration. In practice, we exploit the sparsity of violated rules, so computing the terms $\Psi_{q, MB(i)}$ can be reduced to a constant $O(1)$ time complexity. The computation of the expectation term $E_{y_i \mid MB}\left[\Psi_{q, MB(i)}\right]$ can be estimated using Monte Carlo integration, thus its complexity is $O(n_m)$. Therefore, the overall complexity in rule weight updating step is $O(b_w*(n_m+constant))$ = $O(b_w*n_m)$.
>
> We also empirically observe from experiments, that the rule weight update is quite efficient and most of the run-time consumption comes from the inference step. Since the inference step has the same big O run-time complexity as the training of pure data-driven KG embedding, DiffLogic can be trained efficiently and can be further accelerated by using a larger batch size on a larger GPU.

---

> > ### Comment · Reviewer_wvKd · 2023-08-16
> > **Reply to rebuttal**
> >
> > Thanks for the authors' effort in providing the rebuttal. This helped clarify my concerns. I am happy to keep my current rating.

---

### Official Review · Reviewer_kYuQ · 2023-07-07

**Soundness:** 3 good
**Presentation:** 3 good
**Contribution:** 3 good
**Rating:** 6
**Confidence:** 4

**Summary:**

The paper presents a novel approach for the knowledge base completion task by combining embedding-based and rule-based approaches in a neural symbolic framework. The rule-based component uses probabilistic soft logic to encode rule truth values as continuous values. The overall framework follows an EM approach, where rule weight updating and embedding updating are alternately happening. Later, the authors showed better performance compared to baseline models for the task of knowledge base completion. Additionally, the paper conducted analyses on the efficiency of learning rules and their weight, as well as the effectiveness of injecting new/unseen rules into the framework.


**Strengths:**

1. The proposed approach effectively combines the strengths of both embedding-based and rule-based approaches, resulting in a comprehensive framework. The method is well-motivated, and the empirical results of the proposed approach are strong compared to previous baselines.
2. The rule injection analysis demonstrates the efficacy of injecting additional new rules into the framework. This experiment highlights the flexibility and adaptability of the proposed approach.
3. The additional rule violation analysis is strong compared to the base knowledge base completion model RotatE.

**Weaknesses:**

1. There should be an analysis of the weight parameter between the rule module and the embedding model at inference time. Understanding the behavior would enhance understanding of the model's inference process: whether it relies on the rule system or the embedding system. It's also interesting to know whether changing this parameter would result in a big performance difference.
2. Similarly, it would be great to see the rule weight change before and after learning. Evaluating the changes in rule weights during the learning process would provide valuable information about the evolution of the rules and their influence on the final model performance.


**Questions:**

1. The initialization method for the knowledge graph (KG) embeddings is random, right? Is there a performance change if you try to initialize the embeddings using pre-trained KB models?
2. In the middle of Figure 3, the training MRR and test MRR are almost the same to each. Typically, there is a difference between the two metrics. Is this happening with all the models? Are the train and eval data so similar to each other? Did you try to overfit the training data and see how the test results changed?

**Limitations:**

yes, they discussed the limitations of the methods with relying on high quality rules.

---

> ### Author Rebuttal · Authors · 2023-08-09
>
> Thank you for your questions. We will first answer your questions, and then address the weakness.
>
> Q1. The initialization method for the knowledge graph (KG) embeddings is random, right? Is there a performance change if you try to initialize the embeddings using pre-trained KB models?
>
>
> By default, the KG embeddings are randomly initialized. By changing the initialization method, the final performance does not change as the optimization objective is not changed. However, by initializing the embeddings with pre-trained embeddings (e.g., a RotatE model pre-trained using margin loss and negative sampling), it will take fewer epochs to converge.
>
> {\color{red}Q2. In the middle of Figure 3, the training MRR and test MRR are almost the same to each. Typically, there is a difference between the two metrics. Is this happening with all the models? Are the train and eval data so similar to each other? Did you try to overfit the training data and see how the test results changed?}
>
> Thank you for your question.
>
> Allow us to clarify the confusion. In fact, the two almost overlapped MRR curves in Figure 3 are both training MRR, where one curve is for DiffLogic (red, solid line) and the other is for RotatE (green, dotted line), so the two curves overlap does not mean the training and testing accuracy coincide.
>
> The similarities between these MRR evolution curves stem from three main factors:
>
> 1) We adopt RotatE as the KG embedding model of DiffLogic.
>
> 2) During inference (or embedding updating), the learning objectives of DiffLogic are a composite of the objective of RotatE and the rule-based objective.
>
> 3) Both curves represent training MRRs, indicating how well the RotatE model is fitted to the training data.
>
> Response to the two weaknesses.
>
> Thank you for your kind advice. The rule weights in DiffLogic are dynamically updated during the rule weight learning step, and then the updated rule weights are used to enhance the embedding learning step. We will include more analysis of the rule module in our manuscripts. Thanks for your suggestions!

---

> > ### Comment · Reviewer_kYuQ · 2023-08-15
> > **Thanks for answering my questions!**
> >
> > I read the response and will keep my score.

---

### Official Review · Reviewer_Djat · 2023-07-07

**Soundness:** 3 good
**Presentation:** 4 excellent
**Contribution:** 3 good
**Rating:** 7
**Confidence:** 2

**Summary:**

This paper introduces differentiable logic approach, DiffLogic, based on the probabilistic soft logic (PSL) representation.  Efficient training of DiffLogic is enabled through the introduction of a grounding technique that iteratively identifies important ground formulas required for inference, and additionally develops a  fast estimation technique for computing the gradient of rule weights.  The authors demonstrate this on a subset of common benchmarking tasks commonly used in the field.

**Strengths:**

1. The paper is well written and clear in its goals.
2. Originality of the work resides in the specific embedding representation, which is interesting and straight forward; relevant derivations are presented clearly in the appendix.


**Weaknesses:**

1.  The significance of the work is difficult to assess, as the authors do not compare to many other differentiable nuerual logic approaches that have appeared of the past 5 years, either quantitatively or conceptually?
2. While the title claims to apply to Large-Scale Knowledge Graphs, the authors do not evaluate on one of the most common such datasets, FB15k-237, one of the larger common evaluation datasets used in this field.  It is worth note that the CoDEx-L dataset is somewhat comparable, but not as common in the relevant literature.
3. When comparing evaluations on CoDEx-L, the authors do not include TuckER performance; TuckER was one of the baselines in the original CoDEx paper and appears to outperform the method presented in this paper.

**Questions:**

Can you explain this method and its performance in the context of recent similar methods?
This method seems somewhat complicated to reproduce.  Is there a software implementation available?

**Limitations:**

The authors could better address the distinction between logical and purely embedding representations, incorporate observations about embedding methods that outperform the differentiable logic approach, and note the significant advantages of a differentiable logic method relative to a pure GNN.

---

> ### Author Rebuttal · Authors · 2023-08-09
>
> Thank you for your question. We will first answer the question raised by the reviewer which concerns the first weakness, then we will respond to weakness 2 and weakness 3.
>
> Q1. Can you explain this method and its performance in the context of recent similar methods? This method seems somewhat complicated to reproduce. Is there a software implementation available?
>
> A.
> Our proposed method, DiffLogic, is a neuro-symbolic approach that combines Markov Logic Networks (MLN) with neural methods. It takes advantage of MLN’s ability to inject knowledge encoded in rules and leverages neural methods’ ability to learn from data. The distinguishing factor of DiffLogic lies in its continuous nature, which brings advantages in terms of optimization efficiency and performance.
>
> Recent similar neuro-symbolic methods such as pLogicNet and ExpressGNN also attempt to combine the advantages of MLN and neural methods. Here's how DiffLogic compares to these:
>
> 1) Optimization: Both pLogicNet and ExpressGNN use a neural model to approximate the discrete assignments $x$ and $y$ in an MLN. Finding the optimal approximation is computationally costly because it involves variational inference over a large space. These models avoid this computation by optimizing an Evidence Lower Bound (ELBO) instead. On the contrary, DiffLogic, due to its continuous nature, can directly optimize the MLN objective, thereby offering better computational efficiency and better consistency with rules.
>
> 2) Experimental setting: ExpressGNN requires querying the test dataset during inference, which limits its ability to generalize. On the contrary, DiffLogic only requires the training and validation set during training. Therefore, once DiffLogic is trained, it can generalize to unseen data without any further training, providing it with a superior generalization ability.
>
> Regarding the implementation, we have implemented DiffLogic in Python. We intend to open-source the official code upon acceptance of this paper. Thank you for showing interest!
>
> Response to weakness 2.
>
> Thank you for your kind suggestion. We didn't include fb15k-237 because the datasets used in our experiments are already large and challenging. Specifically, the number of training triples in Codex-l and YAGO3-10 is 551K and 1.08M, respectively, larger than fb15k-237 which contains 272K training triples. We will add the results for fb15k-237 in our manuscripts, and  thanks for your advice.
>
> Response to weakness 3.
>
> In our original manuscript, we use the negative sampling scheme as uniform negative sampling during our implementation of all KG embedding models for fair comparison. If we use adversarial negative sampling instead, our model's performance on Codex-l achieve higher performance (MRR=0.337 and Hit@10=0.46) than TuckER (MRR=0.309, HIT@10=0.430) and other baselines. We will add the comparision with Tucker in our revision. Thanks for the suggestion.
>
> I hope this address your concerns about our experiment results.

---

> > ### Comment · Reviewer_Djat · 2023-08-15
> >
> > Thank you for your explanation and taking the time to answer my questions.   I concur with your response to weakness section item 2, indeed it would be fine to relegate fb15k-237 results to supplementary material.
> >
> > For item 3, your technique's improved performance from (MRR=0.284 and Hit@10=0.412) to (MRR=0.337 and Hit@10=0.46) now outperforms TuckER, but also suggests that the method deserves a more complete answer than you've provided to item 1.
> >
> > Weighing the results, I have revised my assessment.

---

> > > ### Author Response · Authors · 2023-08-15
> > >
> > > Dear reviewer,
> > >
> > > Thank you for your support!
> > >
> > > We sincerely appreciate the time you committed to providing us with constructive feedback.

---

### Official Review · Reviewer_KuwS · 2023-07-09

**Soundness:** 2 fair
**Presentation:** 2 fair
**Contribution:** 2 fair
**Rating:** 6
**Confidence:** 3

**Summary:**

This paper proposed a differentiable framework - DiffLogic. Instead of directly approximating all possible triples, the author design a tailored filter to adaptively select essential triples based on the dynamic rules and weights. The truth scores assessed by KG-embedding are continuous, so the author employ a continuous Markov logic network named probabilistic soft logic (PSL). It employs the truth scores of essential triples to assess the overall agreement among rules, weights, and observed triples.

**Strengths:**

(1) This paper develops a unified neuro-symbolic framework — DiffLogic, that combines the advantages of KG-embedding models and rule-based models: efficiency, effectiveness, capability of leveraging  prior knowledge and handling uncertainty.

(2) This paper enables consistent training of rule-based models and KG-embedding models. By employing PSL, the joint probability of truth scores can be optimized directly rather than optimizing an evidence lower bound (ELBO) instead.

(3) This paper proposes an efficient grounding technique that iteratively identifies important ground formulas required for inference, enabling effective and data-efficient optimization.


**Weaknesses:**

The most impressing advantage of Neuro-Symbolic methods is they are interpretable. But I do not see that.

What is the application of this reasoning method?

**Questions:**

See above.

**Limitations:**

See above.

---

> ### Author Rebuttal · Authors · 2023-08-09
>
> Q1. The most impressing advantage of Neuro-Symbolic methods is they are interpretable. But I do not see that.
>
> A.
> The interpretability of neuro-symbolic methods primarily stems from their logic formulas, using rules to infer new facts is interpretable because a rule is an implication from premise to condition. Take for example, the rule $Father(A, B) \wedge Wife(C, A) \Rightarrow Mother(C, B)$. Inferring a new fact using this rule becomes interpretable when we assign roles to individuals such as A=Jack, B=Ross, C=Judy, and have relations “Father(Jack, Ross)” and “Wife(Judy, Jack)”. Consequently, the conclusion “Mother(Judy, Ross)” is obtained in an interpretable manner.
>
> In DiffLogic, we utilize the logic rules mined by external rule-mining systems. These rules are employed to infer new facts from existing ones in knowledge graphs. Moreover, our rule weight updating step enables learning the importance scores of each rule, which reflects the accuracy of these rules. Therefore enable the results to be interpreted with rules and also demonstrate the confidence of the inference with a score.
>
> During inference in the final step, both embedding scores and rule scores are combined to perform inference. In the scenario where interpretability is needed, DiffLogic can extract ground formulas connected to the inferred facts to interpret the results.
>
> Q2. What is the application of this reasoning method?
>
> A.
> DiffLogic can be employed to perform accurate and interpretable knowledge graph reasoning. We can take advantage of its reasoning ability and interpretability for downstream tasks such as:
>
> 1. Personalized Recommendation Systems: These systems use knowledge graphs (or user-item interaction graphs) to deliver personalized suggestions relevant to a user's preferences, history and behavior.
>
> 2. Question Answering Systems: Digital assistants like Google Assistant, Alexa, utilize knowledge graphs to comprehend and respond accurately to complex questions.
>
> 3. Entity Linking and Disambiguation: Recognizing entities within a text and linking them to corresponding entities in the knowledge graph. Further, it helps decide which entity an ambiguous term refers to, given the context.

---

> > ### Comment · Reviewer_KuwS · 2023-08-17
> >
> > Thanks for your detailed response to my questions. Most of my concerns are addressed.

---

### Official Review · Reviewer_jUBS · 2023-07-16

**Soundness:** 2 fair
**Presentation:** 3 good
**Contribution:** 2 fair
**Rating:** 6
**Confidence:** 3

**Summary:**

The paper proposes a framework called DiffLogic for neuro-symbolic reasoning on knowledge graphs. It balances accuracy and efficiency by selecting essential triples based on dynamic rules and weights. The framework uses a continuous Markov logic network named probabilistic soft logic (PSL) for end-to-end differentiable optimization. Empirical results show that DiffLogic outperforms baselines in both effectiveness and efficiency.

**Strengths:**

1. The idea taking advantage of both MLN and embedding methods makes sense.

2. There is little typo.

3. Several formulas are provided to describe the proposed method.

**Weaknesses:**

1. The authors only classify the KG reasoning methods into two kinds. But there are many methods using graph neural network in recent years for KG reasoning.

2. The EM-algorithm is a common practice in MLN-based methods. It is better for the authors to add some discussion in terms of difference between DiffLogic, pLogicNet and MLN4KB. It seems that MLN4KB is an important and relevant baseline, but there lacks discussion in Section 1, 2, 5.

3. Most of the compared methods are out of date (before 2020, except MLN4KB). Important baselines like RNNLogic[1] and RLogic[2] are missing. ExpressGNN, although cited, is not compared in the experiments.


4. Some of the results in Table 1 are problematic.
- For RotatE, the performance on YAGO3-10 is (MRR=0.495, hit@10=0.670) in their appendix.
- For DRUM, the performance on WN18RR is (MRR=0.486, hit@10=0.586).
I did not check the other values, but considering the results above, the improvement is not significant. Based on my experience, several methods have MRR>0.5 on YAGO3-10 recently.

5. The scalability analysis in Section 4.2 is based on a small data Kinship. I think it should be better to show the running time, memory cost on larger KGs used in this paper to support scalability claims.

6. In Section 4.3, the analysis is based on an unused dataset WN18. It is quite strange that the authors use different sets of datasets in different parts.


[1] RNNLogic: Learning Logic Rules for Reasoning on Knowledge Graphs. ICLR 2021

[2] RLogic: Recursive Logical Rule Learning from Knowledge Graphs. KDD 2022

**Questions:**


1. Can you provide a thorougher literature review?
2. Can you provide a detailed discussion between the MLN-based methods?
3. Can you explain the results in Table 1?
4. Can you explain the usage of datasets?


**Limitations:**

This paper does not include contents discussing limitations and impacts.

---

> ### Author Rebuttal · Authors · 2023-08-09
>
> Q1.
> Can you provide a more thorough literature review?
>
> A.
> Thanks for your suggestion.
> We will do a more thorough literature review, including recent GNN-based methods. We will also add RNNLogic and RLogic as baseline methods in our experiments. See Q3 for more details.
>
> Q2.
> Can you provide a detailed discussion between the MLN-based methods?
>
> A.
> Here we compare our model (DiffLogic) with MLN4KB and pLogicNet, respectively.
> DiffLogic leverages logic rules through a continuous and differentiable MLN framework called probabilistic soft logic (PSL), which allows for smooth integration of MLN and KGE models and result in time and space efficient implementation.
> Both MLN4KB and pLogicNet are built on MLN,
> their inference procedure is essentially a discrete optimization problem and requires sophisticated approximation to solve it.
> MLN4KB only uses rules and can not make use of the similarity between entities as KGE models.
> Though pLogicNet also uses KGE, its overall framework is non-differentiable because it needs to annotate new facts by MLN to train the KGE model.
>
> Q3.
> Can you explain the results in Table 1?
>
> A.
> Yes, we will answer your question from two aspects: 1) reliability of baseline results; 2) comparison with other baselines such as RNNLogic and RLogic.
>
> 1) Reliability of baseline results - {weakness 4.1}
>
> The unmatched baseline performance (RotatE MRR/hit@10 on YAGO3-10) is due to our choice of a uniform negative sampling scheme.  KGE models may employ different negative sampling schemes, e.g., TransE employs a uniform sampling and RotatE employs an adversarial negative sampling. To make a fair comparison, we applied a uniform sampling scheme across all models. Below we attach the results of RotatE and our model under adversarial negative sampling, and argue that the choice of negative sampling schemes does not affect the conclusion in our paper.
>
> |CodeX-s MRR|Hit@10|CodeX-m MRR|Hit@10|CodeX-l MRR|Hit@10|WN18RR MRR|Hit@10|YAGO3-10 MRR|Hit@10|
> |-|-|-|-|-|-|-|-|-|-|
> |RotatE|0.421|0.634|0.325|0.466|0.319|0.453|0.469|0.566|0.495|0.670|
> |DiffLogic|0.445|0.662|0.335|0.487|0.326|0.448|0.493|0.585|0.503|0.673|
> |DiffLogic$^+$|0.458|0.655|0.343|0.495|0.337|0.46|0.50|0.587|0.513|0.674|
>
> We can see that the results of RotatE are now matched. Meanwhile, the results for DiffLogic are also improved and still outperform or are comparable to other baselines.
>
> Thanks for your careful review, we will clarify this subtle point and include the comparison with adversarial negative sampling in our manuscript.
>
> 2) Comparison with other baselines -
> {weakness 4.2 and 3}
>
> We present the results of other baselines in the following, including AIME3, DRUM(t=2/t=3), RNNLogic, and RLogic. Note that these baselines are all rule-learning methods.
>
> ||CodeX-s MRR|Hit@10|CodeX-m MRR|Hit@10|CodeX-l MRR|Hit@10|WN18RR MRR|Hit@10|YAGO3-10 MRR|Hit@10|
> |-|-|-|-|-|-|-|-|-|-|-|
> |AMIE3|0.195|0.283|0.063|0.095|0.026|0.029|0.36|0.485|0.25|0.343|
> |DRUM(T=2)|0.290|0.393|NA|NA|NA|NA|0.434|0.565|NA|NA|
> |DRUM(T=3)|0.342|0.542|NA|NA|NA|NA|0.486|0.586|NA|NA|
> |RNNLogic$^+$|-|-|-|-|-|-|0.51|0.597|NA|NA|
> |RLogic$^+$|-|-|-|-|-|-|0.52|0.604|0.53|0.703|
>
> Regarding the comparison with rule-learning methods, we want to highlight that our model uses simple rules (rule body length $\le$ 2) extracted by AMIE3, while DRUM(t=3), RNNLogic, and RLogic are advanced rule learning systems and can extract longer high-quality rules. We kindly argue that an immediate comparison of our methodologies with other advanced rule mining systems could lead to an unfair comparison. If more high-quality rules are used in DiffLogic, the performance can be improved.We will include the results in our revision.
>
> Regarding ExpressGNN, as we point out in lines 245 -247 of the manuscript, it requires querying test data during training which is non-applicable in our experimental setting, thus we excluded it from our baselines. More details can be found in the OpenReview discussion at https://openreview.net/forum?id=rJg76kStwH.
>
> Q4. Can you explain the usage of datasets?
>
> A.
> Yes, we chose the WN18 dataset because it's more challenging and suitable to demonstrate DiffLogic's rule-injection ability.
> Specifically, injecting rule patterns effectively back into the learned embeddings becomes challenging when the number of rules is small. In this situation, WN18 successfully injects the rule pattern into representations with only 7 rules, even though a significant portion (approximately 36\%) of the original training set has been removed. By comparison, other datasets have more high-scoring rules (confidence score $>$ 0.8)
> (YAGO3-10, WN18RR, Codex-S, Codex-M, Codex-L have
> 22, 13, 35, 52, 56 rules, respectively), making them less challenging.
>
> Concerns about scalability.
>
> For weakness 5, we provide the run-time and memory overhead of the rule grounding process on real-world datasets. The run-time is evaluated 10 times to report mean and std.
>
> |Dataset |Grounding run-time(/sec)|Memory overhead(/MB)|
> |-|-|-|
> |YAGO3-10|3.20±0.04|262.65|
> |WN18RR|0.54±0.01|18.73|
> |CodeX-s|0.03±0.00|2.19|
> |CodeX-m|0.38±0.01|11.57|
> |CodeX-l|0.87±0.04|25.58|
>
> We hope our response can satisfactorily address your concerns and we would appreciate if you could consider raising the score.

---

> > ### Comment · Reviewer_jUBS · 2023-08-13
> > **After rebuttal comments**
> >
> > Thanks for your detailed response to my questions. Most of my concerns are addressed and I have updated my rating.

---

> > > ### Author Response · Authors · 2023-08-14
> > >
> > > Dear reviewer,
> > >
> > > Thanks for the support!
> > > We sincerely appreciate the time and effort you have invested in assessing our work, and thank you for providing us with constructive feedback!

---

### Decision · Program_Chairs · 2023-09-21

**Decision:**

Accept (poster)

**Comment:**

To integrates rule-based reasoning and knowledge graph (KG) embedding, The paper shows how to use probabilistic soft logic to assess the agreement between the inferred triples and weighted rules, based on the embedding representations of entities and relations. All reviews agree that this is an interesting and important topic. Still the point our some downsides such as a weak discussion of related work, no discussion of the potential loss of interpretability, and some missing experimental settings that should be discussed.